# Interaction between Myricetin Aggregates and Lipase under Simplified Intestinal Conditions [note 1]

**DOI:** 10.3390/foods9060777

**Published:** 2020-06-11

**Authors:** Atma-Sol Bustos, Andreas Håkansson, Javier A. Linares-Pastén, Lars Nilsson

**Affiliations:** 1Food Technology, Faculty of Engineering LTH, Lund University, PO Box 124, S-221 00 Lund, Sweden; andreas.hakansson@food.lth.se (A.H.); lars.nilsson@food.lth.se (L.N.); 2School of Chemistry, Faculty of Pure and Natural Sciences, Universidad Mayor de San Andrés, PO Box 303, La Paz, Bolivia; 3Biotechnology, Faculty of Engineering LTH, Lund University, PO Box 117, S-221 00 Lund, Sweden; javier.linares-pasten@biotek.lu.se

**Keywords:** myricetin aggregates, lipase inhibition, intestinal conditions, sequestering mechanism

## Abstract

Myricetin, a flavonoid found in the plant kingdom, has previously been identified as a food molecule with beneficial effects against obesity. This property has been related with its potential to inhibit lipase, the enzyme responsible for fat digestion. In this study, we investigate the interaction between myricetin and lipase under simplified intestinal conditions from a colloidal point of view. The results show that myricetin form aggregates in aqueous medium and under simplified intestinal condition, where it was found that lipase is in its monomeric form. Although lipase inhibition by myricetin at a molecular level has been reported previously, the results of this study suggest that myricetin aggregates inhibit lipase by a sequestering mechanism as well. The size of these aggregates was determined to be in the range of a few nm to >200 nm.

## 1. Introduction

Myricetin is a polyphenol member of the flavonoids family. It is commonly found in the plant kingdom and present in a diversity of foods such as berries, teas, and wines [1,2]. Besides its antioxidant properties, myricetin has been claimed to display several beneficial activities such as analgesic, anti-inflammatory, antitumor, and antidiabetic activities [3,4,5]. As other compounds of the same family, myricetin can inhibit several digestive enzymes [6], for example, lipase, the main enzyme in fat digestion. This enzyme has been intensely studied because of its relation to antiobesity treatments [7,8]. In addition to myricetin, some other flavonoids have been proposed to be lipase inhibitors, for example, it has been shown that quercetin and flavonoids from black tea can bind to lipase near the active site [9,10]. Many of these studies are based on molecular docking, where the flavonoids are considered soluble compounds and the interaction happens at the molecular level [9,10,11]. On the other hand, other studies have shown that some flavonoids can aggregate in aqueous media and inhibit proteins by a sequestering mechanism [12,13].

The myricetin content in some edible products is higher than its aqueous solubility, for instance, green tea has been reported to contain approximately 6 μg of myricetin per ml [14], whereas the aqueous solubility is <1.5 μg/mL [15]. Thus, it is likely that the majority of the molecules are not present as dissolved compounds and that they could be present as, for instance, supramolecular aggregates. The presence of myricetin aggregates in solution can influence enzymatic assays. Bustos et al. (2019) have shown that the presence of phenolic aggregates can affect the reproducibility of lipase assays. Pohjala et al. (2012) have also discussed this issue and suggested that this disturbance can be reduced by adding surfactants that reduce aggregate formation when using enzymatic assays in the presence of flavonoids.

A challenge to applications trying to harness the many beneficial effects of myricetin is its low bioavailability (10% absolute oral bioavailability in rats) [16]. In order to enhance its oral bioavailability, different approaches have been considered such as self-nanoemulsifying drug delivery systems [17] and myricetin co-crystal formation [15]. However, there is a lack of information about pure myricetin aggregates. We argue that a fundamental colloidal understanding of myricetin aggregates in aqueous solutions and their interaction with digestive enzymes can help to understand its beneficial effects.

In this study, we investigate myricetin under in vitro intestinal conditions and its interaction with pancreatic lipase since myricetin is absorbed in the small intestine [18]. The hypothesis of this study is that myricetin can form aggregates under intestinal conditions and can inhibit lipase by a sequestering mechanism.

One of the gentlest techniques to analyze and separate proteins and aggregates is asymmetrical flow field-flow fractionation (AF4), where the separation of analytes is based on their diffusion coefficient and, thus, hydrodynamic radius (r_h_) [19]. As compared to size exclusion chromatography (SEC), no stationary phase is utilized in AF4, thus reducing the potential loss of analyte by adsorption and the breakdown of aggregates by shear forces. Depending on the coupled detectors, AF4 can provide different properties of the analyte, such as size and molecular weight among others. The characterization and separation of proteins, protein oligomers, and higher aggregates with AF4 is, by now, well established [20]. Thus, this technique is suitable for the research questions in this work.

## 2. Methodology

### 2.1. Chemical Compounds

Pancreatic lipase from porcine pancreas Type II (30–90 units/mg protein using triacetin), myricetin (PubChem CID: 5280343), Trizma base (PubChem CID: 6503), and bovine serum albumin (BSA) were purchased from Sigma Aldrich (St. Louis, MO, USA). Dimethyl sulphoxide (PubChem CID: 679) was purchased from VWR Chemicals (Fontenay-sous-Bois, France). All chemicals had purity >95%.

### 2.2. Sample Preparation

In order to simulate intestinal conditions, the standardized static in vitro digestion method proposed by INFOGEST [21] was used as a guide. For this, three different stock solutions were prepared: (1) simulated intestinal fluid (SIF) electrolyte solution (as described by INFOGEST), (2) CaCl_2_ 3 mM solution that is added separately to the SIF solution to prevent precipitation, and (3) saturated lipase solution prepared by dissolving 10 mg of pancreatic lipase in 1 mL of SIF stock solution, followed by centrifugation for 10 min at room temperature at 11,000 g. The lipase stock solution had a final concentration of 20 µm, determined using BCA protein assay Kit (Thermo Scientific) with bovine serum albumin (BSA) as reference protein. In addition, 7 different myricetin stock solutions were prepared in DMSO and mixed with the other solutions to get the next final concentrations: 0, 40, 60, 200, 400, 600, and 1000 µm. The final intestinal solution was prepared, as shown in Table 1. This solution is based on a “typical example” for the intestinal phase proposed by INFOGEST, where bile salts were replaced with water and only lipase was used as enzyme in order to study mainly lipase effects. In addition, aqueous solutions were prepared for each myricetin concentration together with two kinds of control samples (Table 1). The final DMSO concentration in solution was 0.07 mM; this value is in the range of typical lipase activity assays [22,23].

### 2.3. Aggregate Formation

The aggregate formation in myricetin–water and myricetin–intestinal solution samples was measured by turbidity. For this, optical density (OD) changes, for four different myricetin concentrations (60, 200, 600, and 1000 µm) and a blank (0 µm), were measured at 800 nm with a microplate reader (Spectrostar Nano with MARS 3.20 R2 data analysis software, BMG Labtech, Germany). An increase in optical density, with respect to the blank, was taken to indicate aggregate formation. The aggregation experiments were performed at and 37 °C for 2 h (standard digestion time in the INFOGEST method). All measurements were performed in duplicates.

### 2.4. Myricetin–Lipase Interaction

#### 2.4.1. Sample Treatment

In order to study the interaction between myricetin and lipase, the remaining lipase in solution was determined after its interaction with different concentrations of myricetin. For this purpose, eight samples were prepared following Table 1 (see Section 2.2 Sample preparation): one myricetin–control sample of 1000 µm that contained only myricetin and not lipase; one lipase–control sample (0 µm myricetin concentration); and six different myricetin–intestinal solutions: 40, 60, 200, 400, 600, and 1000 µm. After 2 h of incubation at room temperature, the samples were centrifuged at 11,000 g for 10 min. The supernatant was filtered with a filter syringe with cut-off at 0.2 µm (VWR International, USA) before the analysis. All the samples were analyzed by AF4 with multiple detectors, see description in the next subsection.

#### 2.4.2. AF4 Instrumentation

The AF4 system was an Eclipse 3+ (Wyatt Technology, Dernbach, Germany) connected to a UV detector operating at 330 nm (UV-975 detector, Jasco Corp., Tokyo, Japan), to a multiangle light scattering (MALS) detector with a wavelength of 663.8 nm (Dawn Heleos II, Wyatt Technology) and a differential refractive index (dRI) detector (Optilab T–rEX, Wyatt Technology) operating at 658.0 nm wavelength. An Agilent 1100 pump (Agilent Technologies, Waldbronn, Germany) coupled to a vacuum degasser was used to deliver the carrier liquid. The injection of the sample onto the channel was performed by an Agilent 1100 auto-sampler. For the analysis, a trapezoidal long channel (Wyatt Technology) with 26.0 cm length and inlet and outlet widths of 2.15 and 0.6 cm, respectively, was used. The nominal channel thickness was 350 μm. An ultrafiltration membrane of regenerated cellulose was used for the accumulation wall, with 10 kDa nominal cut-off (Merck Millipore, Bedford, MA, USA). 2 mg/mL BSA solution was used to verify the performance of the channel, to normalize the MALS detector and aligning detectors.

#### 2.4.3. AF4 Method Parameters

Prior to the injection, 1 min of elution and 1 min of focus mode were applied to flush and stabilize the channel. The liquid carrier was 20 mM tris-HCl buffer (pH 8) during the whole experiment. 50 μL of sample were injected onto the channel at 0.2 mL/min flow rate for 2 min in focusing mode. After that, 3 min of focusing was applied, followed by 20 min of elution at 5 mL/min constant cross flow followed by 7 min without cross flow to flush the channel. 1 mL/min of detector flow was applied in all the steps.

#### 2.4.4. AF4 Data Processing

Astra software 6.1 (Wyatt Technology) was used for the data analysis. The molecular weight of lipase–control sample was obtained from MALS and dRI detectors, applying the Zimm model [24] with a first order fit using 12 scattering angles (from 44.8° to 149.0°). The refractive index increment (dn/dc) used was 0.185 mL/mg [25], a generic protein value based on BSA. The second virial coefficient was assumed to be negligible.

dRI fractograms were used for lipase quantification, where the peak maximum was used to calculate the relative concentrations, normalized in relation to the lipase-control sample.

The Stokes–Einstein equation was applied to determine the hydrodynamic radius (r_h_),
(1)$$r_{h}=\frac{kT}{6\pi \eta D}$$
where k is the Boltzmann constant, T is the absolute temperature, ƞ is dynamic viscosity of the solvent, and D is the translational diffusion coefficient, using the FFFhydRad 2.1 MATLAB App [26]. The channel thickness (w) was 286.4 µm, determined using BSA with a hydrodynamic diameter of 6.6 nm. The void time (t^0^) was calculated according to Wahlund and Nilsson, 2012.

### 2.5. Statistical Analysis

*t*-tests were used assuming equal variance between conditions. The significance limit was set to 1%.

### 2.6. Molecular Dynamic Simulations

A single monomeric subunit of porcine pancreatic lipase was prepared from the crystallographic structure available in the Protein Data Bank (PDB 1ETH). Atomic coordinates of colipase and other ligands were removed. This monomer was subjected to molecular dynamic simulations in GROMACS 2016 [27] using the AMBER03 force field [28]. The conditions were 20 °C, pH 7.5, and 1 bar, during 500 ns. The solvent was simulated with TIP3P water molecules and sodium chloride ions in a concentration of 0.9% (*w*/*v*) filled into a simulation cubic box of 10 Å extension from the protein. Periodic boundaries, 2.5 fs time steps, and 8 Å cutoff of short-range electrostatic and van der Waals forces and long-range forces calculated by PME were applied [29]. The system was subjected to energy minimization by steepest descent algorithm with a maximum of 50,000 steps considering a step size of 0,1 Å and a tolerance of 1000 kj/mol. Next, a two steps equilibration was performed. First, the temperature was stabilized under NVT ensemble with temperature coupling by a modified Berendsen thermostat [30]. Second, the pressure was stabilized under NPT ensemble with a pressure coupling by the Parrinello–Rahman method [31]. In both steps, the simulation time was 100 ps with time steps of 2 fs. Finally, the production phase was simulated for 500 ns, trajectories were saved every 1.25 ns, and the radius of gyration (r_g_) was calculated.

#### Calculation of the Hydrodynamic Radius from the Molecular Structure

The average molecular structure of the lipase, resulting from the molecular dynamic simulation, was used to calculate the translational diffusion coefficient with HYDROPRO program (Ortega et al., 2011). The hydrodynamic radius (r_h_) was obtained with the Stokes–Einstein equation (Equation (1)).

## 3. Results and Discussions

### 3.1. Aggregates Formation

In order to understand if myricetin forms aggregates in aqueous medium and under intestinal conditions, the turbidity of the different solutions was measured over two hours, considering that a variation in optical density takes place when particles aggregate and cause turbidity. Figure 1 shows that final turbidity results after two hours, where the blank has been subtracted and the error bars obtained from the pooled standard deviation (see Appendix A). The results are expressed as optical densities.

The increased optical density displayed in Figure 1 shows that myricetin can form aggregates in aqueous solutions, as reported previously [32], and that the optical density increases with increasing myricetin concentration. For the data obtained in water, it is possible to see that for 600 and 1000 µm of myricetin, there is a significant change in the optical density (with respect to the blank), indicating that after two hours myricetin aggregates were detected at those concentrations. The lowest concentrations do not show significant changes in the optical density. On the other hand, under intestinal conditions, significant changes in optical density at all the investigated concentrations are detected. The results show that aggregates are formed in the entire investigated concentration range. The optical density when adding myricetin to water is more than double the optical density in the intestinal solution.

### 3.2. Myricetin—Lipase Interaction

The results from the previous section indicate that myricetin form aggregates under the simulated intestinal solution. In order to understand if these aggregates are a combination of myricetin and lipase or consist of myricetin molecules alone, additional experiments were performed.

Figure 2 shows two typical fractograms, AF4–UV and AF4–dRI, of myricetin and lipase-control samples (see Table 1). The dRI–fractograms (A) show a peak at elution time of 4.8 min that corresponds to lipase with a determined molecular weight of 50 kDa (obtained from AF4–MALS–dRI fractograms, see Appendix A); this corresponds to the molecular weight of monomeric lipase previously reported [11]. The myricetin sample does not present any significant peak from this detector, therefore the UV–detector signal was used in order to analyze myricetin. The UV–fractograms (B) shows two peaks: one at 2 min that corresponds to myricetin and the second at 4.8 min that corresponds to lipase. The injected amount of myricetin and lipase, reported in Figure 2, corresponds to the maximum amount analyzed in this study.

As seen in Figure 2, dRI-fractograms give an adequate signal intensity for lipase, but not for myricetin, while UV–fractograms give a more adequate signal intensity for myricetin. Therefore, the remaining lipase in solution, after interaction with myricetin under the simulated intestinal condition, was quantified using AF4–dRI (see Section 3.2.1), while AF4–UV was used to analyze myricetin aggregates (see Section 3.2.2).

It should be noted that all AF4 separations are carried out with an accumulation wall membrane cut-off of 10 kDa. This causes molecularly dissolved myricetin to exit the AF4 channel through the accumulation wall and lost from the separation. Hence, the analyzed myricetin represents supra-molecular aggregates (discussed further in Section 3.2.2).

#### 3.2.1. Sequestering of Lipase

Figure 3A shows the AF4–dRI fractograms of lipase after the interaction with different myricetin concentrations. The retention time remains the same for all the concentrations tested. The height of the peak from Figure 3A is plotted against myricetin concentrations in Figure 3B. The results at all the concentrations are significantly different from the lipase-control sample (0 µm of myricetin), indicating that myricetin can sequester lipase at all the investigated concentrations. Myricetin can sequester up to 20 % of lipase under simplified intestinal conditions (Figure 3B). As a decrease in lipase concentration also means a decrease in activity, the results from Figure 3 show that lipase inhibition by myricetin can occur at the colloidal level and not only at the molecular level, as has been previously reported [33]. Although there are some studies that have shown that flavonoids, such as quercetin, can sequester enzymes [12], this has not been shown for myricetin.

#### 3.2.2. Aggregate Characterization

The aggregates formed between lipase and myricetin can be divided in two classes: (1) The aggregates that were removed by filtration before the AF4 analysis and (2) the aggregates that remained in the sample. For 1) no more experiments were performed, therefore it can only be concluded that the aggregates would have a radius higher than approximately 100 nm (filter cut-off) and that they could be composed of pure myricetin aggregates as well as myricetin–lipase aggregates since myricetin is shown to sequester lipase. These aggregates will be referred to as large myricetin aggregates. For 2, a more detailed characterization is performed with AF4-UV (small myricetin aggregates).

The results from AF4-UV are presented in Figure 4, where two main populations are observed. The first population corresponds to myricetin aggregates, as was also observed in the control samples (Figure 2), and the amount in this population increases with myricetin concentration. The second population represents an increasing amount of aggregates (UV-signal increases) as well as an in aggregate size (population broadens to longer retention times) with increasing myricetin concentration. In the absence of lipase, myricetin does not present any second peak (Figure 2); therefore, the second peak in Figure 4 should consist of lipase–myricetin aggregates.

The hydrodynamic radii (r_h_) for the small myricetin aggregates (Figure 4), myricetin–lipase aggregates (Figure 4), and for lipase (Figure 2) were estimated from AF4 retention time. In order to estimate the r_h_ with sufficient reliability, a retention level >5 is required [19]. The retention level (R_L_) can be calculated according to Equation (2)
(2)$$R_{L}=\frac{t_{r}}{t^{0}}$$
were t_r_ represents the retention time and t^0^ the void time. Note that for the peaks in Figure 4, R_L_ > 5. The results for the estimated r_h_ as well as the retention levels are given in Table 2. The t^0^ is 19 s.

From Table 2, we can see that the size of the small myricetin aggregates is in the range of a few nm. Large myricetin aggregates were previously detected by dynamic light scattering (DLS) [32,34], but the use of separation with AF4 allowed for the detection of smaller myricetin aggregates without the interference of large aggregates. The myricetin–lipase aggregates are somewhat large but remain in a size range <10 nm.

The AF4-MALS-dRI analysis (Appendix A) shows that the molecular weight of lipase corresponds to its monomeric form (see Section 3.2); therefore, the r_h_ of lipase found in Table 2 represents the monomeric form.

#### 3.2.3. Molecular Dynamic Simulations

An important conclusion from Section 3.2.2 is that r_h_ found for lipase corresponds to the monomeric form (r_h_ = 3.1 nm). Since this influences how it can interact with the myricetin, we believe that verification of this finding is required. In order to confirm this result, the size of monomeric lipase was investigated with molecular dynamics simulation. The average radius of gyration obtained was 2.6 nm (see Appendix A) and the hydrodynamic radius of the average structure 3.3 nm. The last is consistent whit the r_h_ obtained experimentally (Table 2), supporting that the lipase studied under the simplified intestinal conditions is present as monomer. In addition, the myricetin–lipase aggregates reach an r_h_ of 4.2 nm (Table 2), suggesting that monomeric lipase could associate with relatively few myricetin molecules forming small aggregates that remain in solution.

Although the results from this study provide an insight into how phenolic compounds can interact with lipase from a colloidal point of view, the simplified conditions used have their limitations. For instance, in a more complex intestinal model, the presence of bile salts could affect the sequestering mechanism suggested in this study; this is because it has been reported that this kind of mechanism caused by phenolic aggregates could be reverted in the presence of surfactants [34,35]. Another important fact, in complex intestinal models, is the presence of natural substrates such as triglyceride. In this case, studies of substrate affinity should be performed in order to provide results closer to reality.

## 4. Conclusions

In this paper, the interaction and aggregate formation between myricetin and lipase was studied in water and simulated intestinal conditions. Myricetin forms aggregates under both conditions, with a size-range from a few nm to above 100 nm. The extent of aggregate formation is dependent on myricetin concentration. Furthermore, the myricetin aggregates can interact with lipase under simplified intestinal conditions and cause sequestering of lipase from solution. The sequestering, thus, causes a decrease in lipase activity, and the amount of lipase sequestered is dependent on myricetin concentration.

## Figures and Tables

**Figure 1 foods-09-00777-f001:**
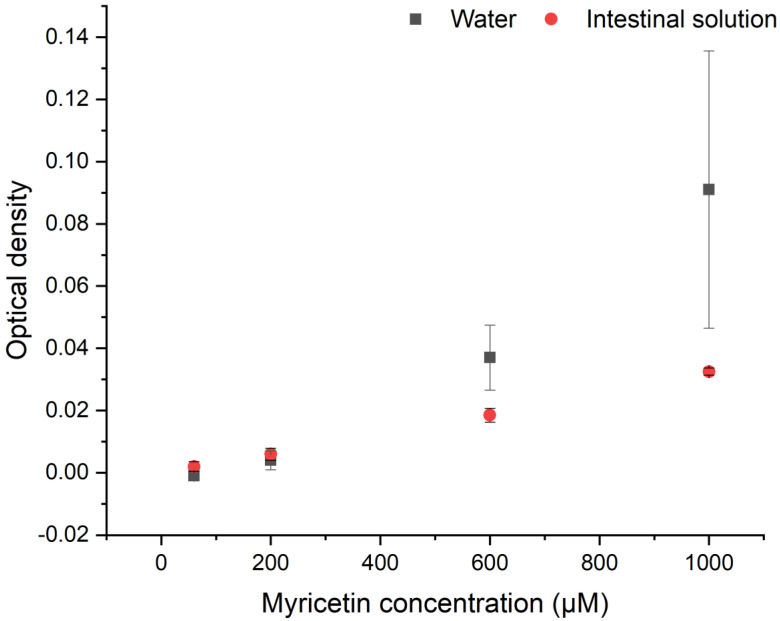
Aggregate formation of myricetin in water and intestinal solution for four different concentrations at 37 °C. The optical density was measured at 37 °C after 2 h of incubation at a wavelength of 800 nm. The error bars represent the pooled standard deviation from duplicates.

**Figure 2 foods-09-00777-f002:**
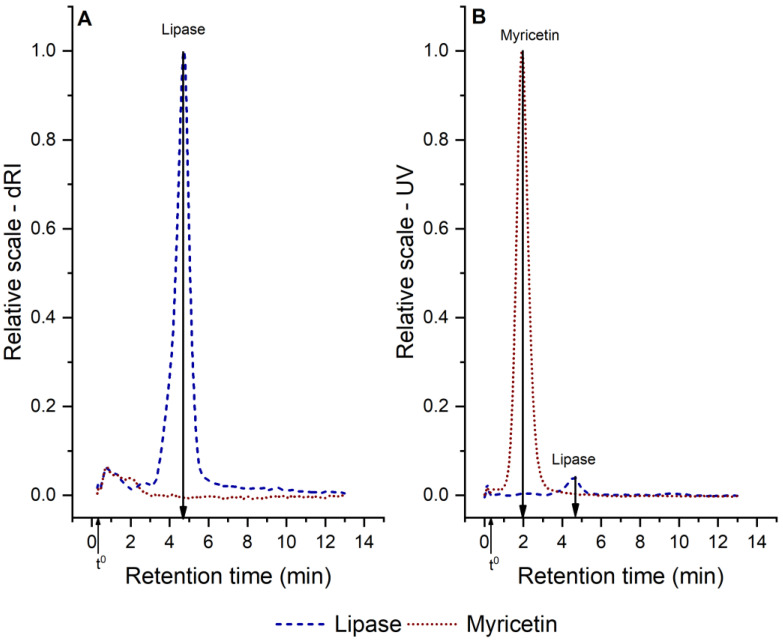
AF4–dRI (**A**) and UV (**B**) fractograms for pure lipase and pure myricetin solution. The injected mass is 25 µg lipase and 16 µg myricetin, respectively. t^0^ denotes the void time at 19 s.

**Figure 3 foods-09-00777-f003:**
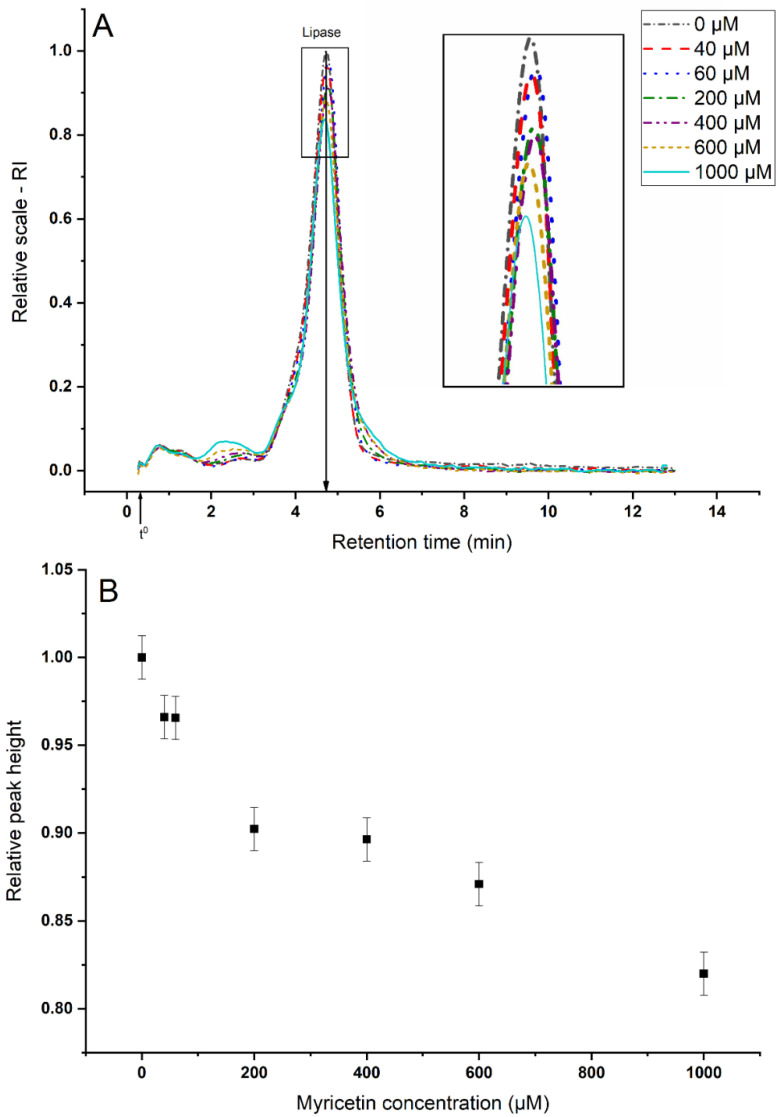
AF4–dRI analyses of the remnant pancreatic lipase after its interaction with different concentrations of myricetin (the sample was filtered before the analyses, see Section 2.4.1). (**A**) Fractograms. (**B**) Relative differential refractive index (dRI) peak heights of lipase vs. myricetin concentration. t^0^ denotes the void time at 19 s.

**Figure 4 foods-09-00777-f004:**
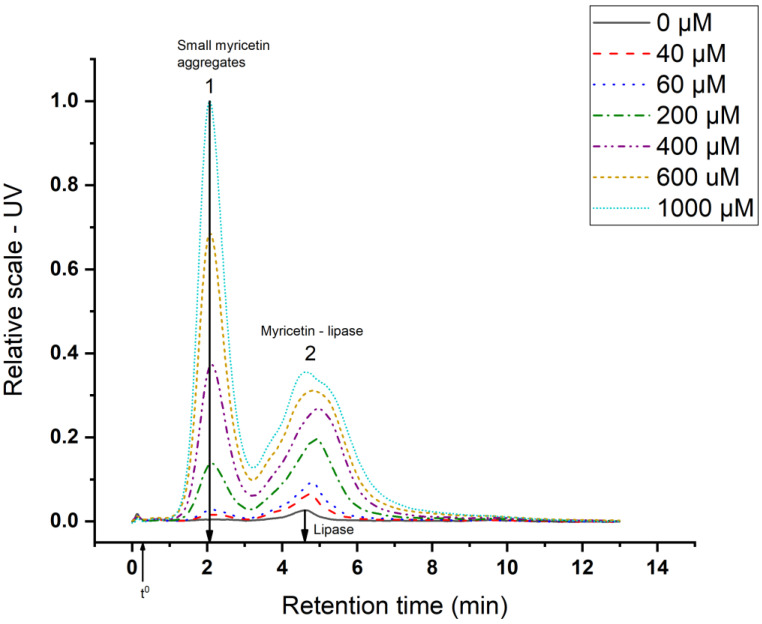
AF4–UV fractogram of different concentrations of myricetin with 0.5 mg/mL lipase. Table 0. sec.

**Table 1 foods-09-00777-t001:** Sample composition.

Sample Name	Water (µL)	SIF Stock (µL)	CaCl_2_ Stock (µL)	Saturated Lipase Stock (µL)	Myricetin-Stock (µL)
Myricetin-intestinal solution	95	0	20	80	5
Myricetin-water	195	-	-	-	5
Myricetin-control	95	80	20	-	5
Lipase-control	95	0	20	80	5 *

* 0 µm myricetin solution.

**Table 2 foods-09-00777-t002:** Hydrodynamic radii.

Analyte	r_h_ (nm)	Retention Time (min)	Retention Level
Lipase	3.1	4.8	15
Small myricetin aggregates	1.3	2.1	7
Myricetin–lipase aggregates	2.2–4.2	3.4–6.5	> 11

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
