# Peer review of "Interaction between Myricetin Aggregates and Lipase under Simplified Intestinal Conditions†"

_foods, 2020, doi:10.3390/foods9060777_

Round 1

Reviewer 1 Report

Interesting study about the binding between flavonoids and proteins. Some explanations are needed

Introduction: should be referred papers about the binding of flavonoids to proteins, there are several on this topic.

Line 176: aggregates, why aggregates and not only turbidity due to the low solubility in water? How is it seen that aggregates are formed?

The raise in absorption is due to higher number of aggregates or higher dimensions of each aggregate?

Line 183: Why is the absorption (optical density) in the intestinal solution lower? Lesser aggregates? Smaller? Which is the pH of the solution? Can’t the pH influence the solubility of myricetin? Can’t protein help the molecule to solubilize by making chemical bonds on its surface, for instance?

Line 211: «Sequestering of lipase»? or flavonoid sequestering by lipase? Flavonoids bind to proteins, can even change their secondary structure, there are several ref on this topic. Why isn’t this aggregation a situation where the flavonoid binds to protein surface? The protein surface can accommodate more than one molecule of flavonoid due to its dimensions. Which is the difference of these myricetin aggregates to protein binding simply?

Line 275: «Myricetin forms aggregates under both solution conditions in a size-range from a few nanometers up to > 100 nm» if it is up to should be 100 nm cannot be >!

General question: if myricetin is aggregated to the enzyme how will it be an inhibitor of its activity

Author Response

Dear reviewer,

We highly appreciate all the comments and we agree that we need to provide more information. Please see all the answers below.

REVIEWER 1

Reviewers' Comments to the Author:

Interesting study about the binding between flavonoids and proteins. Some explanations are needed

Introduction: should be referred papers about the binding of flavonoids to proteins, there are several on this topic.

Response:

Another example of binding between flavonoids and proteins was added in line 34 and two more references related to this topic were added, see lines 35-37.

Line 176: aggregates, why aggregates and not only turbidity due to the low solubility in water? How is it seen that aggregates are formed?

The raise in absorption is due to higher number of aggregates or higher dimensions of each aggregate?

Response:

Considering that a variation in turbidity over time represent aggregate formation, we have added supplementary information about the turbidity results that show not only the turbidity after two hours of incubation, but also the turbidity throughout the two hours, were it is possible to observe a variation over time respect to the blank. Thus, the final turbidity results (after two hours) would represent aggregates. New sentences were added in the original text to clarify this point, see lines 170 – 173.

On the other hand, turbidity studies can provide information about the size or number of aggregates in cases when the particles have a known and defined shape. Therefore, in our case it is hard to know the responsible factor of the raise in absorption and it could be due to a combination of both increase in number and size.

Line 183: Why is the absorption (optical density) in the intestinal solution lower? Lesser aggregates? Smaller? Which is the pH of the solution? Can’t the pH influence the solubility of myricetin? Can’t protein help the molecule to solubilize by making chemical bonds on its surface, for instance?

Response:

The lower optical density would correspond to either smaller aggregates, lesser aggregates or both. The results do at this stage not offer a possibility to make a distinction between the two contributions. The pH of the solution is 7 (INFOGEST method, small intestine), therefore the pH is unlikely to responsible of an improvement in solubility in this situation. In terms of protein, it could potentially bind myricetin and thus have a “solubilizing” effect. Another possibility is that protein adsorb at particle surfaces and prevent colloidal instability leading to large aggregates or prevent further growth of particles. At this stage it is not possible to give an answer to this without an inappropriate amount of speculation. However, we provide some speculation based in our data, see lines 278 - 279.

Line 211: «Sequestering of lipase»? or flavonoid sequestering by lipase? Flavonoids bind to proteins, can even change their secondary structure, there are several ref on this topic. Why isn’t this aggregation a situation where the flavonoid binds to protein surface? The protein surface can accommodate more than one molecule of flavonoid due to its dimensions. Which is the difference of these myricetin aggregates to protein binding simply?

Response:

The title is “Sequestering of lipase”, is not a mistake. This is because the study focus in the amount of lipase after removing the aggregates from the intestinal solution. As you suggest, several possibilities can be behind this phenomena, for example that lipase get attached on the surface of myricetin aggregates, or viceversa, that lipase is covered by myricetin molecules and this new species precipitate or form new aggregates among them. This manuscript is limited to show that the presence of myricetin in the simulated intestinal conditions allows the formation of aggregates that when they are removed from the dispersion, lipase is also removed as part of the aggregates. The detailed mode of aggregation is another research question which is beyond the scope of the current manuscript and is topic of future studies.

Line 275: «Myricetin forms aggregates under both solution conditions in a size-range from a few nanometers up to > 100 nm» if it is up to should be 100 nm cannot be >!

Response:

Thank you for your observation. The conclusion has been modified, see line 283.

General question: if myricetin is aggregated to the enzyme how will it be an inhibitor of its activity

Response:

The results of this manuscript show that myricetin precipitate together with lipase, which would result in a decreasing of the concentration of enzyme that leads to a decrease in enzyme activity.

Reviewer 2 Report

The manuscript aims to evaluate the in vitro effect of myricetin on the pancreatic lipase, with particular focus on the trapping of the enzyme, in order to study a putative inhibition of the enzyme.

The topic of the manuscript is really of interest because it focuses on the putative role of polyphenols in inhibiting the lipase activity, and in turns the fat absorption.

However, I think that, the simplified intestinal system let the results lacking of important considerations.

Authors are studying the effects of the myricetin on lipase, but not considering the lipase activity, and they did not considered the substrate of the lipase. Pancreatic lipase is secerned into the duodenum when the digestion of the lipids occurs, so the best approach would be put in the mix of reaction between myricetin and lipase also the triglyceride solution. In fact, it is possible that when the enzyme is linked to its substrate, the steric hindrance as well as other properties might be different for the myricetin triggering. Plus, usually pancreatic lipase works in presence of bile acids micelle which englobes the fat, and this is another variable for the bond myricetin to lipase. Authors should carefully discuss this point

Can authors determine the concentration of the mother solution of myricetin in DMSO? DMSO should be carefully diluted not to fold the enzyme, and actually myricetin is poorly polar.

Author Response

Dear reviewer,

We highly appreciate all the comments and we agree that we need to provide more information. Please see all the answers below.

REVIEWER 2

Reviewers' Comments to the Author:

The manuscript aims to evaluate the in vitro effect of myricetin on the pancreatic lipase, with particular focus on the trapping of the enzyme, in order to study a putative inhibition of the enzyme.

The topic of the manuscript is really of interest because it focuses on the putative role of polyphenols in inhibiting the lipase activity, and in turns the fat absorption.

However, I think that, the simplified intestinal system let the results lacking of important considerations.

Authors are studying the effects of the myricetin on lipase, but not considering the lipase activity, and they did not considered the substrate of the lipase. Pancreatic lipase is secerned into the duodenum when the digestion of the lipids occurs, so the best approach would be put in the mix of reaction between myricetin and lipase also the triglyceride solution. In fact, it is possible that when the enzyme is linked to its substrate, the steric hindrance as well as other properties might be different for the myricetin triggering. Plus, usually pancreatic lipase works in presence of bile acids micelle which englobes the fat, and this is another variable for the bond myricetin to lipase. Authors should carefully discuss this point

Can authors determine the concentration of the mother solution of myricetin in DMSO? DMSO should be carefully diluted not to fold the enzyme, and actually myricetin is poorly polar.

Response:

The purpose of the experiments reported is to see if interaction would occur at a pH and salt environment which is similar to intestinal conditions. Hence, the terminology “simplified” intestinal conditions is used. This is a first simplified model system to investigate the interaction. The plan would then be to gradually increase the complexity to the full INFOGEST protocol. However, one needs to investigate the role of individual parameters first before studying the entire complex system. The reviewer is correct that for instance substrate and bile acids would be present during the digestion process. Although it is our opinion that other components from the simulated intestinal conditions that are omitted in this exploratory simplified method (e.g. bile salts, enzymes other than lipase, phospholipids etc.) would already be expected to play a role for interaction of myricetin and lipase. Hence, these components should first be included in future work before additional complexity is added (e.g. triglyceride emulsion droplets).

Although there is some speculation about this topic, we prefer to discuss only the current results and leave further conclusions until they can be supported by experimental results. A small comment was added to the manuscript about this observation, see lines 288 - 291.

About the DMSO concentration, the concentration of the mother solution of myricetin in DMSO was 40 mM (1000µM in the final solution). The DMSO concentration in the final solution was 2.5%. Such a low concentration of DMSO would not be expected to have any significant impact on the studied phenomena and therefore we consider the influence of DMSO as negligible.

Round 2

Reviewer 1 Report

OK

Author Response

No comments to be answer 

Reviewer 2 Report

I strongly believe that the model authors are considering in this work cannot describe the in vivo situation of the lipase activity, and authors cannot limit their discussion just stating that future models of increasing complexity might elucidate more aspects. In fact, the model of lipase that authors are considering is not working in the in vivo system, and so cannot describe any particular behavior of myricetin, as other substrates are not participating to the reaction.

2.5% DMSO is still a relevant concentration, by thinking that in cell culture system it is usually used lower than 0.1% because it melts membranes and de-folds protein. So, unless authors proof that lipase is not destroyed by this amount, I really think this is a tricky point worth to be considered.

Author Response

Reviewer 2

Comments and Suggestions for Authors

I strongly believe that the model authors are considering in this work cannot describe the in vivo situation of the lipase activity, and authors cannot limit their discussion just stating that future models of increasing complexity might elucidate more aspects. In fact, the model of lipase that authors are considering is not working in the in vivo system, and so cannot describe any particular behavior of myricetin, as other substrates are not participating to the reaction.

2.5% DMSO is still a relevant concentration, by thinking that in cell culture system it is usually used lower than 0.1% because it melts membranes and de-folds protein. So, unless authors proof that lipase is not destroyed by this amount, I really think this is a tricky point worth to be considered.

Response.

DMSO is considered non-toxic solvent, for instance, its LD50 (14,500 mg/kg) is remarkably higher than ethanol (7,060 mg/kg) as it was tested in rats. Indeed, DMSO is used as ingredient in several pharmacological formulas. These properties make DMSO a suitable co-solvent to be used in simplified intestinal conditions. We disagree with the reviewer; cell culture systems are not comparable with intestinal simplified systems (ISS), therefore the DMSO concentration used in cell cultivation is not valid as reference for ISS. On the other hand, it is very well documented the enzymatic activity of lipases in a broad range of organic solvents, including DMSO (Cao et al. 1992). Indeed, pancreatic lipase reactions are reported in emulsions prepared with DMSO at concentration significantly higher than those used in this work, for instance 6.8 % (Hardvary et al. 1991).

References

Hadvary, P., Sidler, W., Meister, W., Vetter, W., & Wolfer, H. (1991). The lipase inhibitor tetrahydrolipstatin binds covalently to the putative active site serine of pancreatic lipase. Journal of Biological Chemistry, 266(4), 2021-2027.

Cao, S. G., Feng, Y., Liu, Z. B., Ding, Z. T., & Cheng, Y. H. (1992). Lipase catalysis in organic solvents. Applied biochemistry and biotechnology, 32(1-3), 7-13.